

# Diurnal predators of restocked lab-reared and wild *Diadema antillarum* near artificial reefs in Saba

Mareike de Breuyn[1], Alex J. van der Last[1], Oliver J. Klokman[1] and Alwin Hylkema[1,2]

[1] Van Hall Larenstein University of Applied Sciences, Leeuwarden, Friesland, The Netherlands
[2] Marine Animal Ecology group, Wageningen University, Wageningen University & Research, Wageningen, Gelderland, The Netherlands

## ABSTRACT

The long-spined sea urchin *Diadema antillarum* controls reef dynamics by grazing on algae and increasing coral recruitment. Populations of *Diadema* never recovered after a mass-die off in 1983 and 1984, and numbers were further reduced by a more recent die-off in 2022. To restore grazing pressure and thereby the resilience of Caribbean coral reefs, multiple *Diadema* restocking efforts have been performed. Although results vary, relatively low retention is one of the reasons restocking is not considered more often. If causes for the low retention can be identified, suitable measures may be able to increase restocking success. In this study, we monitored restocked lab-reared and wild juvenile *Diadema* on artificial reefs around Saba, Caribbean Netherlands. To assess the retention of *Diadema* over time, we conducted diver surveys and used underwater photo time lapse during daylight. Retention of uncaged lab-reared and wild *Diadema* decreased steadily with less than 30% surviving after 10 days. In total, 138 predator-prey interactions were recorded, of which 99% involved the queen triggerfish *Balistes vetula*, although other potential predators were present in the area. None of the recorded predator-prey interactions was successful, which suggests that artificial reefs with incorporated shelters may be suitable for juveniles as daytime refuge. However, *Diadema* that were more often attacked during the day were more likely to be absent the next morning. Because queen triggerfish often visited the experimental site in the first or last hour of daylight, it could be that they were more successful in their attacks when it was too dark to see anything on the photos and when *Diadema* came out to feed or to look for better shelter opportunities. If *Diadema* migrated off the artificial reef, they were probably predated during the process, because no *Diadema* were found on surrounding reefs. Wild *Diadema* were attacked significantly more often than lab-reared *Diadema*, possibly because the wild urchins were larger, but this did not significantly affect retention. Future restocking should be performed on natural or artificial reefs with deeper shelters, so *Diadema* can retract farther into their crevice, and should include night-time monitoring to identify the remaining unknown factors that cause low retention, including migration and nocturnal predation. This knowledge is urgently needed to coral reef managers so they can increase *Diadema* restocking success by selecting reefs with a lower predator density, protect urchins during an acclimatization period and/or conduct temporary predator control measures.

Corresponding author
Alwin Hylkema,
alwin.hylkema@hvhl.nl

# INTRODUCTION

The long-spined sea urchin *Diadema antillarum*, hereafter *Diadema*, was once an ubiquitous species on Caribbean coral reefs (*Randall, Schroeder & Starck, 1964*; *Sammarco, 1982*; *Bak, Carpay & De Ruyter Van Steveninck, 1984*). It is considered a keystone herbivore as it structures the benthic community through its gregarious grazing behaviour. Between 1983 and 1984, 95–99% of all *Diadema* were killed during one of the most extensive and severe die-offs ever recorded for a marine invertebrate (*Lessios et al., 1984*; *Lessios, Robertson & Cubit, 1984*; *Hughes et al., 1985*; *Hunte, Côté & Tomascik, 1986*; *Levitan, Edmunds & Levitan, 2014*). Without other herbivores to fill the niche (*Mumby et al., 2006*; *Dell et al., 2020*), macroalgae became the dominant benthic group on many Caribbean coral reefs (*Hughes et al., 1985*; *Carpenter, 1986*; *Lessios, 1988*). Other stressors such as disease outbreaks and hurricanes reduced coral cover by as much as 50% (*Hughes, 1994*; *Jackson et al., 2014*; *Cramer et al., 2020*). The emptied space was quickly overgrown by macroalgae and other benthic organisms such as cyanobacteria (*Bakker et al., 2017*) and peyssonnelids (*Williams & García-Sais, 2020*; *Wilson, Fan & Edmunds, 2020*; *Stockton & Edmunds, 2021*), which all inhibit coral recruitment (*Lessios, 1988*; *McCook, Jompa & Diaz-Pulido, 2001*; *Kuffner et al., 2006*). This resulted in coral recruitment failure and a decreased resilience of Caribbean coral reefs (*Bellwood et al., 2004*).

In the decades after the die-off, *Diadema* recovery remained slow. *Lessios (2016)* estimated the *Diadema* density in 2015 as 8.5 times less dense than before the 1983-1984 die-off. The few recovered *Diadema* populations have been linked to reduced macroalgae cover (*Edmunds & Carpenter, 2001*; *Myhre & Acevedo-Gutiérrez, 2007*), increased coral recruitment (*Carpenter & Edmunds, 2006*), survival and growth (*Idjadi, Haring & Precht, 2010*) and ultimately, higher coral cover (*Myhre & Acevedo-Gutiérrez, 2007*). Active restoration of *Diadema* has therefore become a top priority in Caribbean coral reef management (*Bellwood et al., 2004*), especially because a new die-off reduced population densities across the Caribbean in 2022 (*Hylkema et al., 2023*). Approaches to restore *Diadema* include restocking individuals (*Chiappone, Swanson & Miller, 2006*; *Nedimyer & Moe, 2006*; *Dame, 2008*) or 'assisted natural recovery' in which suitable settlement substrate for *Diadema* larvae is supplied on the reef (*Hylkema et al., 2022*). Individuals for restocking can be acquired through culture from gametes (*Pilnick et al., 2021*; *Wijers et al., 2023*) and *in-situ* collection of settlers (*Williams, 2018*; *Williams, 2022*), but most restocking attempts have been performed by translocating individuals from naturally recovered areas to experimental plots (*Chiappone, Swanson & Miller, 2006*; *Nedimyer & Moe, 2006*; *Maciá, Robinson & Nalevanko, 2007*; *Burdick, 2008*; *Dame, 2008*; *Pilnick et al., 2023*).

Some restocking attempts have recorded retention of *Diadema* on experimental reefs of up to 56% after 3 to 12 weeks (*Maciá, Robinson & Nalevanko, 2007*; *Dame, 2008*; *Williams, 2018*; *Pilnick et al., 2023*). However, most restocking attempts had relatively few or no

retained *Diadema* after 3.5 to 12 months (*Chiappone, Swanson & Miller, 2006*; *Nedimyer & Moe, 2006*; *Burdick, 2008*; *De Breuyn, 2021*). Most authors point toward predation (*Chiappone, Swanson & Miller, 2006*; *Nedimyer & Moe, 2006*; *Burdick, 2008*), migration (*Maciá, Robinson & Nalevanko, 2007*; *Williams, 2018*), or a combination of both (*Dame, 2008*; *Wynne, 2008*; *Williams, 2022*) as potential causes for the decline of restocked *Diadema*. Predation may be due to high predation pressure by fishes (*Harborne et al., 2009*), low fitness of lab-reared *Diadema* (*Sharp et al., 2018*) or a lack of available refuges (*Bodmer et al., 2015*; *Pilnick et al., 2023*), while migration may be triggered by low food availability (*Vadas, 1977*) or predator avoidance behaviour (*Snyder & Snyder, 1970*). With the positive effects of recovered *Diadema* populations, the slow recovery in other places, as well as the few successful restocking attempts, the need for the development of successful *Diadema* restocking practices is high and the key factors determining retention must be identified.

On Saba, Caribbean Netherlands, a restocking experiment was conducted with 147 lab-reared juveniles (*De Breuyn, 2021*), which were introduced on artificial reefs with suitable shelters, as recommended by *Delgado & Sharp (2021)*. As with multiple other restocking attempts, retention was low and the cause unknown (*De Breuyn, 2021*). Because spines with tissue chunks, without other urchin remains, were observed by returning researchers as soon as one hour after restocking, the author suggested a diurnal predator as the most important factor affecting retention, but no actual attacks were observed. The aim of the current study was to identify the main predators of restocked *Diadema* on artificial reefs on Saba. We hypothesized that diurnal predation would be the main cause for low retention of *Diadema* at this location. An additional aim of this study was to compare the susceptibility to predation of lab-reared and wild *Diadema*. Individuals from both sources were introduced on standardized artificial reefs and monitored intensively using time lapse cameras. Based on *Sharp et al. (2018)* and *Brundu, Farina & Domenici (2020)*, we hypothesized that lab-reared *Diadema* have a lower retention than wild conspecifics.

## MATERIALS & METHODS

We conducted our field experiments at Big Rock Market (N: 17.36772, W: 063.14264) on the south coast of Saba, Caribbean Netherlands, within the Saba National Marine Park (Fig. 1). Our study site was at a depth of 19 m and near a previous study site, where *Diadema* restocking was unsuccessful due to one or more unidentified predators (*De Breuyn, 2021*).

### Experimental set-up

Moreef (Modular Restoration Reef, http://www.moreef.com) artificial reef modules (height = 50 cm, diameter = 60 cm) were made from concrete in August 2020. Each Moreef module contained eight tapered blind shelters, two tunnel shelters and numerous tapered micro-shelters (Fig. 2). The artificial reefs were deployed in September 2020 and repositioned for the current experiment in March 2021. Twelve Moreef modules were set out in two rows of six on a large sand patch with nearby patch reefs (Fig. 3A). The Moreefs were spaced one meter apart, which was the largest distance which would still allow two reefs to be monitored by a single camera, because only four camera setups were available. The four reefs on the ends of the rows were placed in cages made from chicken wire with a mesh size

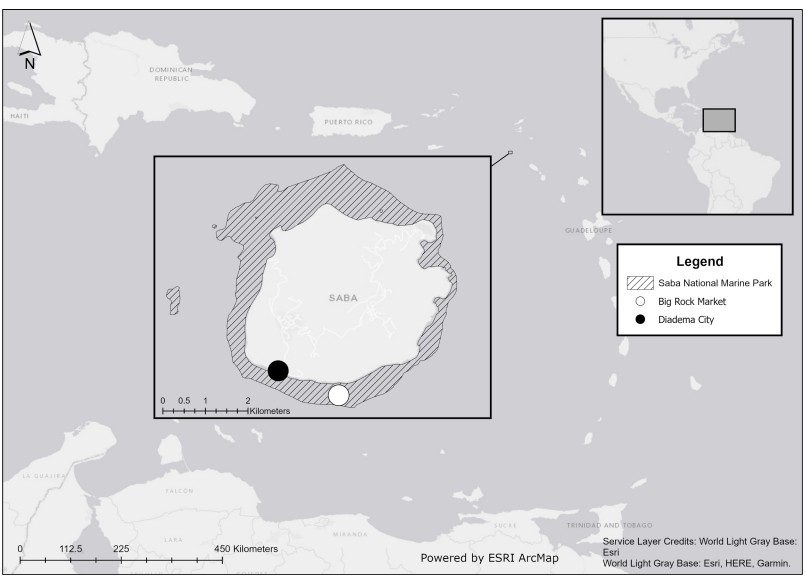

**Figure 1** **Location of Saba in the Caribbean.** Experiments were performed at Big Rock Market (white dot) and wild *Diadema antillarum* were collected at Diadema City (black dot). Map created with ArcMap 10.8 using data from Esri, HERE, and Garmin.

of 1.3 cm as controls to monitor survival when predation and migration were prevented (Fig. 3B).

On 13 April 2021, four *Diadema* were placed in each artificial reef module, one in each blind shelter facing the camera. In total, 48 *Diadema* were introduced. Half were lab-reared and half were wild. Thus, 24 wild and 24 lab-reared *Diadema* were used, with 16 of each type on open modules and 8 of each type on caged control modules. The lab-reared *Diadema* were collected as settlers and head-started in a land-based nursery following the approach of *Williams (2018)*. Wild individuals were collected during the week before the experiment started at the dive site Diadema City (Fig. 1) where a former breakwater harbored the largest population of *Diadema* around Saba at the time of this study. To keep the sizes of wild and lab-reared *Diadema* as similar as possible, we aimed to select wild individuals within the size range of lab-reared *Diadema* from the nursery (17–33 mm test size). However, even when using the smallest collected recruits, the average (±SD) test size of wild individuals was 32.6 ± 5.5 mm, which was larger than the 24.8 ± 4.0 mm size of lab-reared *Diadema*. Permission to relocate *Diadema* for this experiment was given by Kai Wulf from the Saba Conservation Foundation, who was the Saba National Marine Park (SNMP) manager at the time of this study. The SNMP includes the sea and seabed around Saba from the high-water mark to a depth of 60 m.

## Retention surveys and camera set-up

We conducted retention surveys in which divers inspected each shelter for *Diadema* 1, 2, 3, 6, 7 and 10 days after restocking between 08:00 and 09:00. To determine behaviour of *Diadema* and to identify predators, all eight uncaged reefs were monitored with four

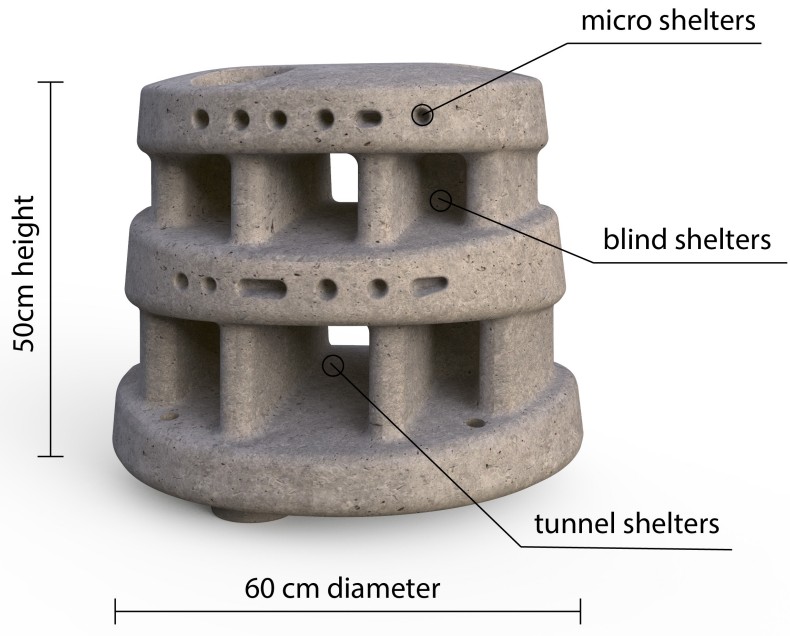

**Figure 2  Moreef artificial reef module.** Front view of Modular Restoration Reef (Moreef) module with incorporated shelters.

underwater camera setups during the 10-day period. Each camera setup consisted of a GoPro 8 (GoPro, Inc.) inside a 10 cm watertight cylinder (Blue Robotics Inc.). Two power banks (V75 USB Battery Pack, Voltaic Systems) with a total capacity of 19,200 mAh per camera were enclosed. The setups were placed on a stand 55 cm above the substrate to which the camera setups could easily be attached and reattached. The cameras setups were installed simultaneously with the introduction of *Diadema* at the start of the experiment (day 0). The setups used a wide-angle setting with a time-lapse interval of 5 s to photograph the blind shelters with introduced *Diadema* and a surrounding margin of one meter to record any activity on the sand and in the surrounding water column. No lights were used, so useable images were restricted to daylight (approximately 05:40 to 18:45). Cameras were removed around midday on days 2 and 5 and replaced on day 3 and day 7 (Fig. S1). At the time of retrieval, the cameras had all stopped because of empty batteries and had run between 32 h and 51 h (average 42 h). This resulted in three deployments each with approximately 20 h of daytime recording over the 10-day period, covering approximately 60% of the daylight hours with more complete coverage in the first half of the experiment.

## Photographic analysis

Four camera setups took photos during three camera installations over 10 days, resulting in approximately 32,400 photos per camera per installation and an overall total of 388,800 photos (four cameras multiplied by three runs). Photos taken within ten minutes after retention surveys or camera deployments were excluded from analysis. Photos taken at night were also excluded as they were entirely black. The remaining 194,400 photos were
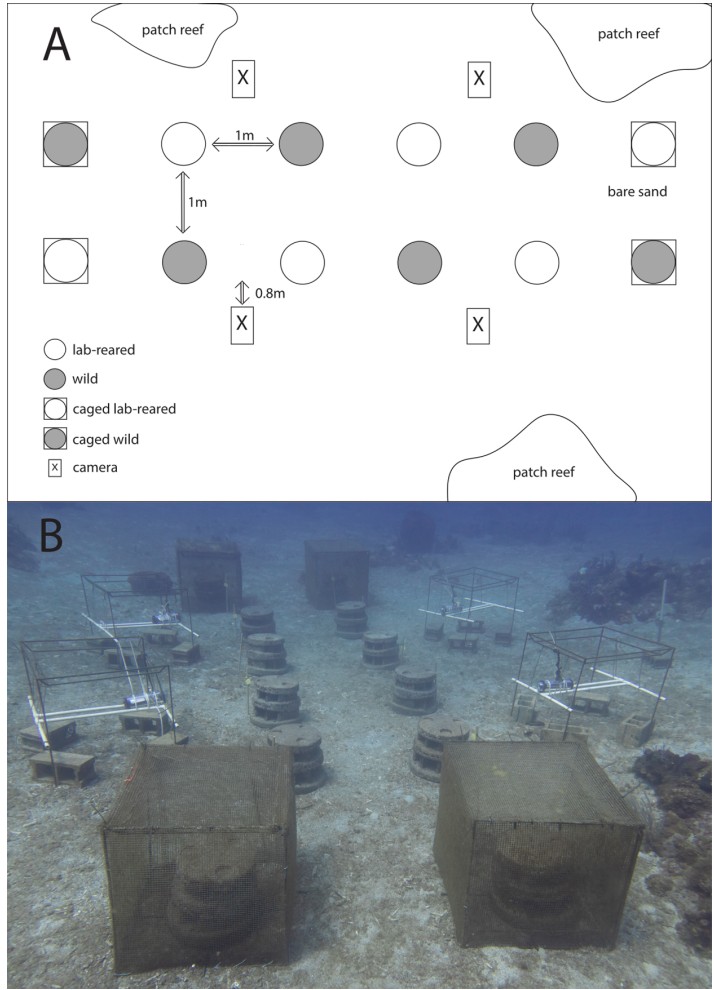

**Figure 3** **Experimental setup.** (A) Schematic overview of the experimental setup. Shown are artificial reefs on bare sand with restocked lab-reared *Diadema antillarum* (white circle) and artificial reefs with restocked wild *Diadema* (grey circle) of which two reefs on each outer end were caged (squared circle). Cameras (square box) were installed to monitor the artificial reefs. Distances in between the artificial reefs are indicated with arrows. (B) Photo of the experimental setup.

manually analysed by MDB and AL. Photos collected during tests of the set-up, including restocked *Diadema*, were analysed by both researchers for training purposes. For analysis, each photo was carefully searched for known predators of *Diadema* and for *Diadema* outside of their shelter space. The list of predators included 13 fish species based on *Randall, Schroeder & Starck (1964)*. These were black margate *Anisotremus surinamensis*, white margate *Haemulon album*, Spanish grunt *Haemulon macrostomum*, Caesar grunt *Haemulon carbonarium*, white grunt *Haemulon plumierii*, bluestriped grunt *Haemulon sciurus*, permit *Trachinotus falcatus*, jolthead porgy *Calamus bajonado*, saucereye porgy *Calamus calamus*, Spanish hogfish *Bodianus rufus*, Caribbean spiny lobster *Panulirus argus*, queen triggerfish *Balistes vetula*, bandtail puffer *Sphoeroides spengleri* and the spotted porcupine fish *Diodon hysterix*. We also included the Caribbean spiny lobster *Panulirus*

*argus* based on *Randall, Schroeder & Starck (1964)* and the spotted spiny lobster *P. guttatus* based on *Kintzing & Butler (2014)*.

Photos were coded according to predefined codes of which examples can be seen in Fig. 4. Predator sightings in individual photos were coded 1–7 and include a code for a predator–prey interaction on the reef (code 4) and off the reef (code 6), as well as a code for a predator feeding on *Diadema* (code 7). Codes 8 and 9 relate to *Diadema* outside of their shelter space in the abscence of a predator. It was not possible to observe attacks on *Diadema* within shelters because they would retreat into the shelter and the predator followed, blocking the view of the cameras. We therefore coded these probable attacks as 'interactions' (code 4) and defined interaction as "photo with predator snout in shelter". Photos were only attributed to the highest level code describing the action. For example, a photo with a predator interacting with *Diadema* in the shelter was only attributed to code 4 and not to code 1, 2 or 3. We installed cameras opposite of each other, so both cameras had two artificial reefs in the front and two in the back of the photo, to account for actions at the back of the artificial reefs. Codes 2–9 were only recorded for the two artificial reefs directly in front of the respective camera, avoiding double counts of the cameras opposite.

## Roving diver survey

To determine the presence of predators on the surrounding reefs, a roving diver visual survey was conducted after completion of the retention count on day 6 when cameras were not running. We based the survey on the fish roving diver technique, which considers presence/absence data as well as frequencies of fish species (*Hill & Wilkinson, 2004*), including only the potential predators listed above. The starting point of the survey was the centre of the experimental plot. Three scuba divers systematically inspected the reefs within a 200 m radius from the experimental plots for 30 min and recorded the size and number of all predators of *Diadema*.

## Statistical analysis

A Generalized Linear Mixed Model (GLMM) was used to assess the effect of source (factor: lab-reared or wild), caging (factor: caged or uncaged) and day of the experiment (covariate) on the retention of *Diadema* per artificial reef (response variable, coded in r as number of urchins retained, number of urchins missing, following *Zuur et al., 2009*). As urchins retained were a proportion of the initial number of restocked individuals, a binomial distribution was used. Models were fit using the glmer function in the R package "lme4" (*Bates et al., 2015*). To account for daily repeated surveys on the same reefs, reef ID was included as random factor. For statistical inference, likelihood ratio tests (LRT) were performed using the drop1 function (*Zuur et al., 2009*). Generalized Linear Models (GLMs) were used to assess the effect of source (fixed factor) on (1) the number of photos on which a predator was within 10 cm of a shelter (code 3) and (2) interacted with the *Diadema* (code 4). Both GLMs were run with artificial reef as replicate, thus using number of photos per artificial reef. Model validation for both models was performed according to *Zuur et al. (2009)*. Initial models were fit with a Poisson distribution (glm function with family = Poisson in the R package "lme4") but turned out to be overdispersed. This

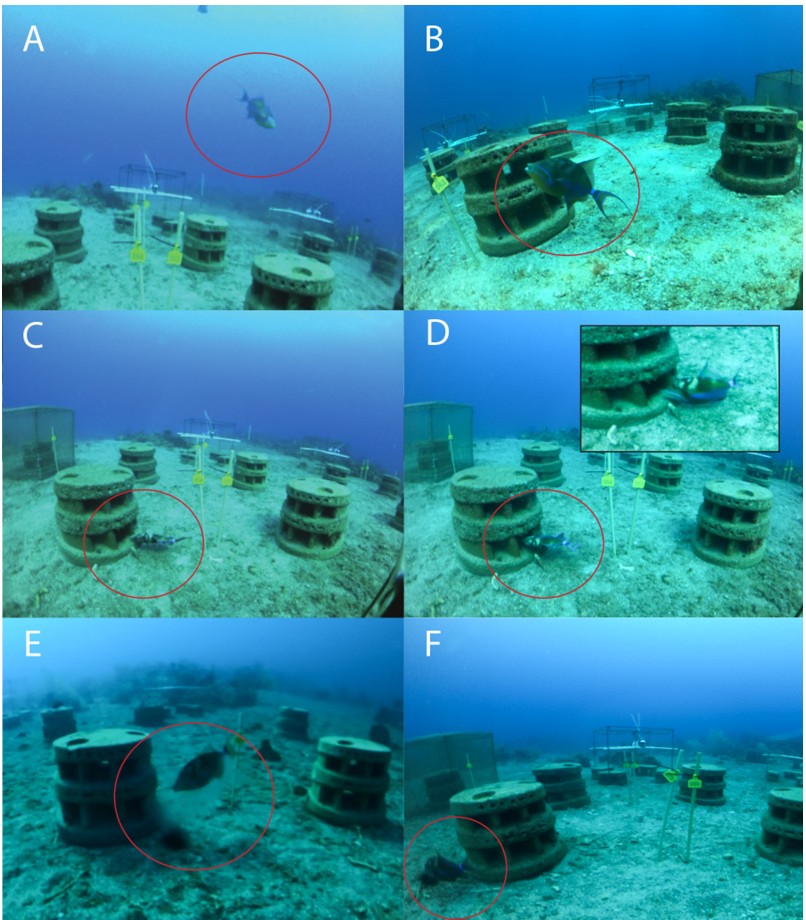

**Figure 4** **Codes to categorize actions of predators and *Diadema antillarum*.** Codes used in this study to categorize actions of predators (circled) and *Diadema antillarum*: (A) Code 1: *Diadema* predator outside a 50 cm virtual sphere around the artificial reef. (B) Code 2: *Diadema* predator less than 50 cm from artificial reef, but less than 10 cm from a shelter entrance. (C) Code 3: *Diadema* predator less than 10 cm from a shelter entrance. (D) Code 4: Interaction between *Diadema* predator and *Diadema* on the artificial reef. (E) Code 5: *Diadema* predator within a 50 cm virtual cylinder around Diadema outside shelter. (F) Code 6: *Diadema* predator attacks *Diadema* outside shelter. Code 7 (*Diadema* predator feeds on *Diadema* outside shelter.), Code 8 (*Diadema* outside shelter and within 50 cm of artificial reef.) and Code 9 (*Diadema* outside shelter and more than 50 cm from the artificial reef. No *Diadema* predator present.) are not shown. Pictures were only attributed to the highest level code describing the action.

was resolved by using a negative-binomial distribution (glm.nb function in the R package "MASS"). Likelihood ratio tests (LRT) were performed for statistical inference of the fixed factors using the drop1 function.

Because photos were taken only 5 s apart, a predator usually appeared in a sequence of multiple photos. We considered visits as independent only if they were separated from other photos of a conspecific predator by at least 10 min. Running time of the cameras was used to calculate the number of independent visits per hour per day. The time of the first and the last photo in a set were used to calculate the duration per independent visit and the mean duration per day.
To test whether the number of interactions at a specific shelter was related to the probability that that shelter would be vacated the next day, a subset of the data was created including only observations of shelters in which there was a single *Diadema* at the start of the first or second night of the experiment. The difference in number of *Diadema* between the start of the night and the next morning was modelled with GLMMs using the glmer function in the R package "lme4". A binomial distribution was used (family = binomial) as the difference in *Diadema* at the beginning and end of the night was either 0 or 1 (presence-absence data). Source and total number of interactions were considered as fixed factors. To account for repeated measures, because the same shelter was surveyed multiple mornings, shelter ID was included as a random factor. Model selection was performed based on AIC (*Zuur et al., 2009*; *Bolker et al., 2009*). For statistical inference, likelihood ratio tests (LRT) were performed using the drop1 function (*Zuur et al., 2009*).

All statistical analyses were performed with R (*R Core Team, 2021*) using RStudio version 1.2.5001 (*RStudio Team, 2019*). *P*-values < 0.05 were considered statistically significant. Reported values are mean ± standard deviation, unless otherwise indicated. The R package "ggplot2" was used to construct the graph.

## RESULTS

Artificial reefs with uncaged wild and lab-reared *Diadema* had, respectively, 31 ± 47% and 25 ± 29% average retention of restocked *Diadema* after 10 days (Fig. 5). All of the caged wild and seven out of eight lab-reared caged *Diadema* survived the experiment. Retention of *Diadema* on the artificial reefs was significantly positively affected by caging (LRT = 13.41, $df = 3$, $P < 0.001$) and significantly negatively related to day of the experiment (LRT = 56.17, $df = 1$, $P < 0.001$). Retention was not significantly affected by source of the sea urchins.

Photo analysis resulted in 648 coded predator photos. All included *Diadema* predators and no sightings were recorded of *Diadema* outside their shelter without a predator present (code 8 and 9). Of all photos with a predator (Table 1), 189 included a predator more than 50 cm from an artificial reef module (code 1), 281 sightings included a predator 10–50 cm of an artificial reef (code 2), 40 sightings included a predator within 10 cm of an artificial reef (code 3), and 136 sightings include interactions between a predator and *Diadema* (code 4). There was a single sighting of a *Diadema* outside its shelter, on the sand, with a predator within 50 cm (code 5) and another single sighting of a predator attacking that same individual (code 6). No sightings were observed of a predator feeding on *Diadema* (code 7). Queen triggerfish was by far the most abundant predator with 589 recorded photos, followed by the spotted porcupine fish with 23 photos, the Caribbean spiny lobster with 22 photos and the Spanish hogfish which was recorded on 11 photos. The bandtail puffer was recorded on two photos and the saucereye porgy was recorded on one photo. Of all predators, only queen triggerfish and Spanish hogfish approached within 10 cm (code 3 and 4). For the Spanish hogfish this was recorded twice, while the other 176 photos concerned the queen triggerfish. Most of these sightings (135) concerned interactions between the queen triggerfish and *Diadema*. On average, 6.0 ± 4.1 photos
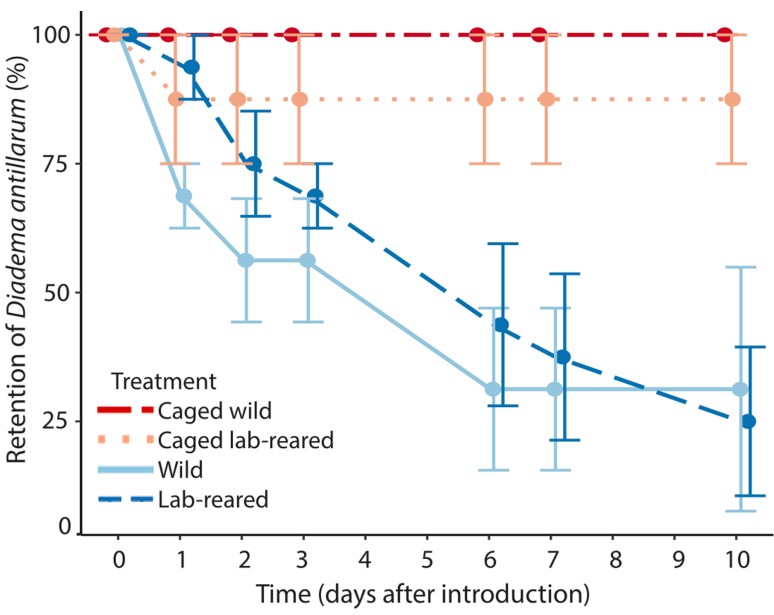

**Figure 5** *Diadema antillarum* **retention.** Mean (±SE) *Diadema antillarum* retention on artificial reefs over the 10-day experiment. Artificial reefs had the following treatments: caged wild *Diadema* (red circles, dot-dash line, $n = 2$), caged lab-reared *Diadema* (light pink circles, dotted line, $n = 2$), uncaged wild *Diadema* (light blue circles, solid line, $n = 4$) and uncaged lab-reared *Diadema* (dark blue circles, dashed line, $n = 4$).

of queen triggerfish within 10 cm of a shelter were recorded per artificial reef restocked with wild *Diadema*. This was not significantly different (LRT = 1.38, $df = 1$, $P = 0.240$) from reefs restocked with lab-reared *Diadema*, where the queen triggerfish was recorded within 10 cm of a shelter on 3.8 ± 2.2 photos per artificial reef module. Interactions of the queen triggerfish with *Diadema* were observed significantly more often on reefs restocked with wild compared to lab-reared *Diadema* (LRT = 11.72, $df = 1$, $P < 0.001$). In total, we recorded 26.2 ± 15.8 photos per artificial reef with interactions between queen triggerfish and wild *Diadema*, and 7.5 ± 2.7 between queen triggerfish and lab-reared *Diadema*.

In total, 104 independent predator visits were recorded, of which 82 concerned the queen triggerfish. Queen triggerfish visits were more frequent in the first half of the experiment, where 1.0–1.8 visits per hour were recorded (Table 2). In the second half of the experiment, this decreased till 0.4–1.0 visits per hour. Mean duration per queen triggerfish visit varied among days and was highest on the first day (11 ± 15 minute) and lowest on day 8 (0 ± 1 minute). On all days with running cameras, first independent queen triggerfish visits were recorded in the first hour of daylight (between 5:40 and 6:40). On four of the six days with running cameras in the afternoon, the last queen triggerfish visit was recorded in the final hour of daylight (between 17:45 and 18:45). Besides the queen triggerfish visits, 22 independent visits from other predators were recorded. Most of these visits lasted less than a minute and concerned predators passing by the experimental site. Only the Spanish hogfish had a single visit that lasted for 12 min.

**Table 1  Diadema antillarum predator photos.** Overview of all photos (n) including a *Diadema antillarum* predator, categorized per code, per predator species and in total. Predator species are sorted based on their number of sightings.

| Common name | Scientific name | Potential predator >50 cm from artificial reef | Potential predator 10–50 cm from artificial reef | Potential predator <10 cm from artificial reef | Interaction predator and *Diadema* | Potential predator <50 cm of *Diadema* outside shelter | Potential predator attacks *Diadema* outside shelter | Potential predator feeds on *Diadema* | Total photos per species: |
|---|---|---|---|---|---|---|---|---|---|
| | Code | 1 | 2 | 3 | 4 | 5 | 6 | 7 | |
| queen triggerfish | *Balistes vetula* | 159 | 254 | 39 | 135 | 1 | 1 | 0 | **589** |
| porcupine fish | *Diodon hysterix* | 20 | 3 | 0 | 0 | 0 | 0 | 0 | **23** |
| Caribbean spiny lobster | *Panulirus argus* | 7 | 15 | 0 | 0 | 0 | 0 | 0 | **22** |
| Spanish hogfish | *Bodianus rufus* | 0 | 9 | 1 | 1 | 0 | 0 | 0 | **11** |
| bandtail pufferfish | *Sphoeroides spengleri* | 2 | 0 | 0 | 0 | 0 | 0 | 0 | **2** |
| saucereye porgy | *Calamus calamus* | 1 | 0 | 0 | 0 | 0 | 0 | 0 | **1** |
| **Total photos per code:** | | **189** | **281** | **40** | **136** | **1** | **1** | **0** | **648** |

**Table 2  Independent queen triggerfish visits.** Number of independent queen triggerfish visits (*n*), mean duration (±SD) per independent visit (h:mm), start time of the first visit (hh:mm), and end time of the last visit (hh:mm) per day.

| Day of experiment | Mean independent visits per hour | Mean duration per independent visit (±SD) (h:mm) | Start time first visit (hh:mm) | End time last visit (hh:mm) |
|---|---|---|---|---|
| 0 | 1.8 | 0:11 ± 0:15 | 12:50[*] | 18:21 |
| 1 | 1.5 | 0:04 ± 0:06 | 05:42 | 18:11 |
| 2 | 1.0 | 0:02 ± 0:03 | 06:20 | 10:22[*] |
| 3 | 1.4 | 0:06 ± 0:07 | 11:06[*] | 17:35 |
| 4 | 1.1 | 0:01 ± 0:02 | 06:21 | 18:28 |
| 5 | 1.0 | 0:04 ± 0:08 | 05:48 | 11:29[*] |
| 6 | nd | nd | nd | nd |
| 7 | 0.4 | 0:05 ± 0:03 | 14:17[*] | 15:09 |
| 8 | 0.7 | 0:00 ± 0:01 | 06:24 | 17:37 |
| 9 | 0.6 | 0:01 ± 0:03 | 05:59 | 13:13[*] |

**Notes.**
[*]Time was affected by camera deployment or retrieval.

Total number of interactions during the day on a particular shelter had a significant relationship to the retention of *Diadema* in that shelter during the following night (LRT = 8.36, $df = 1$, $P = 0.004$). Shelters that retained a *Diadema* at the end of the night ($n = 54$) had 0.93 ± 1.52 interactions with predators during the previous day whereas shelters that

lost their *Diadema* during the night ($n = 22$) had $3.48 \pm 6.20$ interactions with predators. Source had no significant effect on retention and was not included in the best fitting model.

Six *Diadema* predator species were sighted during the roving diver survey. The Caesar grunt was the most abundant with four sightings, followed by two sightings of the spotted spiny lobster. The black margate, Caribbean spiny lobster, queen triggerfish and Spanish hogfish were all sighted once.

## DISCUSSION

Retention of *Diadema* on the artificial reefs was relatively low, falling to 25–30% by 10 days. This was expected, as the current study is a follow-up on a restocking attempt at a nearby location, where a restocking experiment resulted in a mean retention of 0% after 3 months (*De Breuyn, 2021*). The sharp decline in *Diadema* in less than two weeks in the current study makes it unlikely that any of the restocked individuals would have remained on the artificial reefs for longer than a few months. Almost all caged lab-reared and wild *Diadema* survived for the full duration of the experiment, indicating that potential stressors related to the transportation (*e.g.*, changes in oxygen, salinity, and temperature) or handling were of minor concern and that other factors negatively affected retention. Retention of restocked *Diadema* is thought to be mediated by predation pressure, habitat, food availability, and behavioural tendencies (*Miller et al., 2007*; *Keller & Donahue, 2006*; *Williams, 2022*).

Based on the removal of some of the *Diadema* within hours after restocking during a previous experiment (*De Breuyn, 2021*), we hypothesized that diurnal predation would be the major factor affecting retention. Contrary to this hypothesis, no *Diadema* predation was recorded in this study. We did, however, observe many predator–prey interactions, of which the majority were conducted by queen triggerfish, which is known as one of the most important diurnal predators of *Diadema* (*Randall, Schroeder & Starck, 1964*; *Randall, 1968*; *Manooch III & Drennon, 1987*).

Next to queen triggerfish, many other fishes and crustaceans are known as predators of *Diadema* (*Randall, Schroeder & Starck, 1964*; *Kintzing & Butler, 2014*). Of those, spotted porcupine fish, Spanish hogfish and the Caribbean spiny lobster were recorded on more than 10 photos. Only the Spanish hogfish was recorded two times close to the shelter entrance and one of these photos concerned an interaction. In addition to the predators recorded on photos, black margate, Caesar grunt and the spotted spiny lobster were recorded on the adjacent reefs during a roving diver survey. Apparently, most of the predators observed on photos and during the roving diver survey were not attracted by the presence of *Diadema*. This may be an effect of the continued low local densities of *Diadema*, which could have resulted in dietary shifts of certain predators (*Reinthal, Kensley & Lewis, 1984*). The reefs surrounding the experimental site had very low *Diadema* densities, with no individuals observed during this study (personal observations of all authors) suggesting that *Diadema* do not form a significant dietary proportion of predators in the area. More generalist predators such as the wrasses and grunts could therefore be less attracted by low densities of *Diadema*. More specialized predators, such as the queen triggerfish, were able to persist after the 1983–1984 *Diadema* die-off by switching to other prey items in

the absence of their primary prey (*Reinthal, Kensley & Lewis, 1984*), but might still prefer *Diadema*.

The low success of predation attempts indicates that the shelter of the Moreef modules provided suitable protection for *Diadema* during the day. The photos of the interactions indicate that the shelters were too narrow for the snout of queen triggerfish to reach *Diadema* at the deep end of the crevice. *Dame (2008)* conducted a restocking experiment with *Diadema* around Curaçao and concluded that the shape of the shelter affects retention. Both types of shelter tested by *Dame (2008)* showed a decrease in retention throughout the 3-week observation period, but the persistence of *Diadema* was significantly higher in "tunnel" shelters than in "hut" shelters, which had 0% retention after 16 days.

The explanation for the low retention of *Diadema* in this study has to be sought in processes happening at night. *Diadema* usually leave their shelter at dusk to feed (*Randall, Schroeder & Starck, 1964*), which probably made them more vulnerable to predation. Of the predators that were present on the surrounding reefs, spotted porcupinefish (*Carpenter, 1984*), Caribbean spiny lobster (*Lozano-Alvarez & Spanier, 1997*), spotted spiny lobster (*Kintzing & Butler, 2014*) and black margate (*McClanahan, 1999*) are known to be nocturnal and could have preyed upon *Diadema* inside or outside their shelters. Predation during dawn or dusk by queen triggerfish can also not be excluded. The earliest recorded visit of queen triggerfish was around sunrise, at 05:42 and the latest was around sunset, at 18:28. Almost on all days of the experiment there were queen triggerfish visits during the first hour of daylight, the last hour of daylight, or both. It could therefore be that queen triggerfish preyed upon *Diadema* when it was too dark to see anything on the photos and we therefore could no record the successful attacks. As the interactions during the day indicate that the shelters provided sufficient protection against the queen triggerfish, it is likely that, if this hypothesis is true, *Diadema* had left their shelter voluntarily. The correlation between shelters that had a lot of interactions during the day and shelters that were vacated during the following night, support the hypothesis that queen triggerfish preyed upon the *Diadema*, because it is likely that these fish had a preference for the same specimens during dawn or dusk as during the day.

However, another explanation for the correlation could be that *Diadema* migrated off the artificial reef to look for better shelter. *Diadema* can assess the quality of their shelter and will more readily vacate poorer quality shelters under simulated predation (*Carpenter, 1984*), which likely occurred in the present study and explains why shelters that were attacked more often had a lower retention rate. Other restocking studies also hypothesized that habitat features were a driver of losses in retention (*Miller et al., 2007*; *Keller & Donahue, 2006*). Small test reefs (*Miller et al., 2007*; *Levitan & Genovese, 1989*) and limited reef complexity (*Keller & Donahue, 2006*; *Dame, 2008*; *Pilnick et al., 2023*) were possible explanations for migration. Although not part of our study design, we opportunistically inspected the surrounding reefs for *Diadema* at the time of the experiment. Like *Miller et al. (2007)* and contrary to *Dame (2008)* and *Williams (2016)*, not a single *Diadema* was found. Although it is entirely possible that some of the *Diadema* were overlooked while hiding in the natural reef, it is unlikely we missed them all. This suggests that migration, if it occurred, was interrupted by predation during the night. Individual *Diadema* on sand

have little protection (*Levitan & Genovese, 1989*), which could be an explanation why these individuals were not found on the natural reefs.

Another incentive for *Diadema* migration is to find conspecifics to aggregate with. This is a known defence mechanism of *Diadema* (*Kintzing & Butler, 2014*) and has been experimentally shown to increase juvenile survival (*Miller et al., 2007*). The limited size of the artificial reefs used in this study did not allow large *Diadema* aggregations and could have been a reason for migration off the artificial reef. *Diadema* could also have moved off the artificial reefs to find food elsewhere. Although this alternative hypothesis cannot be totally disregarded, the artificial reefs were well overgrown with turf algae and some macroalgae, which reduces the chance that *Diadema* were migrating off the artificial reefs in search of food. Nevertheless, causation of post-translocation movements remains poorly understood, and attempts to stock reefs with higher densities of adults (*Wynne, 2008*) and on high rugosity reefs (*Keller & Donahue, 2006*) still resulted in migration, even if predation remained low.

Contrary to our hypothesis, wild *Diadema* were attacked significantly more often compared to lab-reared individuals. This was unexpected, because lab-reared *Diadema* can exhibit reduced diel sheltering behaviour, which would increase vulnerability to predation, compared to wild urchins (*Sharp et al., 2018*). Nonetheless, in our study, no *Diadema* were recorded outside their shelter during the day and both lab-reared and wild *Diadema* were sheltering towards the back of the shelters. The lack of a difference in sheltering behaviour between lab-reared and wild urchins can potentially be accounted for by the lab-reared urchins having grown under normal day-night rhythms and in semi-rugose aquaria, as recommended by *Sharp et al. (2018)*, *Sharp et al. (2023)* and *Hassan et al. (2022)*. In addition, the high number of unsuccessful predator–prey interactions during the day likely provided increased stimulus to shelter (*Carpenter, 1984*). A final explanation for the higher number of interactions on wild *Diadema* is that they were larger compared to the lab-reared urchins. Possibly, queen triggerfish prefers larger prey or it could be that larger prey is simply more readily detected or easier to attack, as they will be easier to reach when residing at the back of the shelter. The larger size of the wild *Diadema* made it possible for the researchers to easily distinguish lab-reared from wild *Diadema* during retention counts. No wild *Diadema* were found on artificial reefs that were supposed to have lab-reared *Diadema* and vice versa (personal observation MDB and AL), excluding the possibility that the source treatment became mixed-up by nocturnal movements of the *Diadema*. The higher number of interactions with wild *Diadema* did not affect the final retention, which was similar for both sources.

## CONCLUSIONS

We conclude that the low retention of *Diadema* during the present study is likely a result of predation and/or migration at night. There are multiple possible explanations for this, but *Diadema* that were more often attacked by queen triggerfish during the day were more likely to be missing the next morning. It could be that queen triggerfish were more successful in their attacks when it was too dark to see anything on the photos and when

*Diadema* came out to feed or to look for better shelter opportunities. *Diadema* are known to look for better shelter when attacked, so migration off the artificial reef could still be the result of interactions with queen triggerfish during the day. If this was indeed the case, these *Diadema* were probably predated during the process, because no *Diadema* were found on surrounding reefs. No indications were found that lab-reared individuals were less suitable than wild *Diadema* for restocking practices, although it cannot be ruled out that lab-reared individuals were initially attacked less because of their smaller size. To increase restocking success, future restocking attempts should be conducted on artificial or natural reefs that have shelters more than 20 cm deep, so *Diadema* can retreat far enough to avoid contact with predators. We recommend monitoring restocked *Diadema* also at night and at other locations, to determine the causative factors for low *Diadema* retention, including identification of the most important predators. This information is essential to give coral reef managers the opportunity to increase *Diadema* restocking success by selecting reefs with a lower predator density, giving restocked *Diadema* an acclimatization period in a protected environment (*Williams, 2022*), and/or conducting temporary predator control measures. Since Caribbean coral reefs continue to degrade and a new die-off reduced *Diadema* densities in large parts of the Caribbean in 2022 (*Hylkema et al., 2023*), the development of effective restocking practices is urgently needed.

## ACKNOWLEDGEMENTS

We would like to thank the Saba Conservation Foundation for providing us with a workspace to conduct this project; with a special thanks to Walter Hynds Dilbert, Ayumi Kuramae Izioka, Tom Brokke and Marijn van der Laan for their assistance. We further thank Esther van de Pas and Marnik Lehwald for help during the experimental setup and camera installations. We greatly appreciate the suggestions of Patrick Bron and Jorien Rippen on the research proposal. We are grateful for the excellent reviews from Dr. Donald Kramer, Dr. Joshua Patterson, and an anonymous reviewer, which helped us improve an earlier version of this manuscript.

### Funding

This research was conducted in the context of the RAAK PRO Diadema project (project# RAAK.PRO03.005), partly funded by SIA, part of the Dutch Research Council (NWO). The funders had no role in study design, data collection and analysis, decision to publish, or preparation of the manuscript.

### Grant Disclosures

The following grant information was disclosed by the authors:
RAAK PRO Diadema project: RAAK.PRO03.005.
SIA.

## Competing Interests

Alwin Hylkema is co-founder of Moreef, a company producing the molds for the Modular Restoration Reef (Moreef) used in this study. The function of the Moreef was not studied and the Moreefs only functioned as a standardized habitat for the sea urchins.

## Author Contributions

- Mareike de Breuyn conceived and designed the experiments, performed the experiments, analyzed the data, prepared figures and/or tables, authored or reviewed drafts of the article, and approved the final draft.
- Alex J. van der Last conceived and designed the experiments, performed the experiments, analyzed the data, prepared figures and/or tables, authored or reviewed drafts of the article, and approved the final draft.
- Oliver J. Klokman conceived and designed the experiments, authored or reviewed drafts of the article, and approved the final draft.
- Alwin Hylkema conceived and designed the experiments, analyzed the data, prepared figures and/or tables, authored or reviewed drafts of the article, and approved the final draft.

## Field Study Permissions

The following information was supplied relating to field study approvals (*i.e.*, approving body and any reference numbers):

Permission for this study was given by the Saba Conservation Foundation (SCF).

## Data Availability

The raw data are available in the Supplementary File.

## Supplemental Information

Supplemental information for this article can be found online at http://dx.doi.org/10.7717/peerj.16189#supplemental-information.

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
