# Peer review of "Diurnal predators of restocked lab-reared and wild Diadema antillarum near artificial reefs in Saba"

_PeerJ, doi:10.7717/peerj.16189_

## Round 0.1 · original submission · Minor Revisions

Overview
This study used underwater time-lapse photography of small artificial reef modules stocked with wild-caught or lab-raised juvenile long-spined sea urchins to attempt to document predation and emigration as potential causes of the typically poor retention of re-stocked individuals. Cameras operated for about 75% of the duration of the 10-day study and only recorded during daylight hours. Retention was low with less than 30% of re-stocked individuals surviving 10 days and no significant differences between wild or lab-raised urchins; there was very little mortality in controls caged to prevent predation and emigration. No actual predation was observed, but 136 photos out of nearly 200,000 recorded predation attempts by queen triggerfish and another 510 recorded the presence of potential predators, nearly all queen triggerfish, in the area. It was not clear how many separate visits to the experimental site by the predator(s) may have been involved. Only one photo recorded an urchin outside its shelter on the reef module. However, urchins with more photos of attempted predation during one day were more likely to be missing from their shelter the following morning. It was not clear whether the larger number of photos represented more frequent or longer duration visits by predators and whether the loss of urchins was due to nocturnal predation in the shelters or to emigration, potentially followed by predation of exposed individuals. Attempted predation occurred more often with the wild-caught than with the lab-raised urchins. The authors suggest that shelters on artificial reef modules should be deeper to provide better protection and that nocturnal observations are needed.

Both reviewers reported that the experiment was sound, although the sample size was limited, and had no comments on the statistical analysis. They indicated that the quality of the presentation was generally good but suggested some improvements and areas needing clarification. Reviewer 1 provided some additional references and indicated areas where the conclusions need to be strengthened. I also enjoyed reading the manuscript but had a few major concerns and more minor ones and many suggestions for grammatical corrections and clearer, more concise writing. Many of my minor suggestions or questions were made directly on the pdf. You may treat my comments as a third review, i.e., make changes if appropriate or explain why you have elected not to make changes. You do not need to reply to all the grammatical and style comments on the pdf unless you disagree.

Editor’s Comments

Title: The title suggests a causal relationship between attempted predation and retention failure. However, the pattern is correlational and might have other explanations (see my comments below). Furthermore, this title alerts readers to only one aspect of your findings. Consider a less causal title such as ‘Diurnal attacks by queen triggerfish predict overnight loss of long-spined sea urchins restocked on artificial reefs’ or a more general title such as ‘Diurnal predators of long-spined sea urchins near restocked artificial reefs in Saba’.

Names. There is no problem using D. antillarum to refer to the species throughout the manuscript after providing the common name in the Abstract and first mention in the Introduction. However, you might find it simpler to write ‘hereafter, urchins’ at this first mention and then use ‘urchins’ throughout the manuscript. For the predators, again provide common and scientific names at first use and then only one throughout. You may have repeated the combined names later in the manuscript. My personal preference would be for the common names because many readers will not be able to remember the list of scientific names, but either approach is acceptable.

L63 (and elsewhere). I don’t see a need to refer to ‘remote’ photos. This term refers to cameras triggered from a distance whereas yours were automatic. I don’t think the camera trap literature uses this term. Please check whether the term adds relevant information.

Key Words: consider adding the specific name of the artificial reef modules and the families of triggerfish and urchins.

Introduction
L126. In the references, de Breuyn (2021) is listed as ‘unpublished data’. If this is correct, i.e. only data, it should not be in the references, but ‘de Breuyn 2021 unpublished data’ could be cited in parentheses in the text. If it is an actual report, that readers could request, the citation needs to be more complete and refer to an unpublished report rather than unpublished data. Also, is there a discrepancy between the in text citation of de Breuyn but the reference as Breuyn, M de. Non-Dutch readers would look for de Breuyn under D, not B. However, if this is proper Dutch citation form, this minor confusion is no problem.

Methods
I have proposed a number of changes to the Methods to make a more logical and complete development of the information. I made the number of individuals per treatment more explicit because I had some difficulty figuring it out. For the information about the time period covered by the photographs, I made some assumptions about deployment times and dawn and dusk. You should correct as needed, but the information does seem to be important.
L151. Check Instructions to Authors. I think you are supposed to use capital letters for panels in a multi-panel figure.
L175. Given that the diameter of the wild urchins was about 33% larger than the lab-raised, and that volume increases as the cube of a linear measure, are you really justified in calling these ‘slightly larger’ (also in Abstract)?
L199. Check use of the term ‘picture’. I think most authors refer to image analysis or photo analysis. Change other references to pictures throughout the text.
L200. ‘Throughout’ implies continuous which does not apply to your study.
L200ff. For the photo analysis, I felt there was a need for some clarifications.
• I calculated that you should have had about 259,200 photos rather than the 194,400 referred to (two 12-h recordings plus one 6-h recording per deployment = 30 h x 60 min/h x 12 photos/min x 3 deployments x 4 cameras. The discrepancy can’t be explained simply by the 10 min exclusions for installment and checking retention but might be explained by the time of dawn or dusk or the time at which light levels were sufficient for useable photos (should be specified) or by the record not starting right at dawn (should be specified). It is important to indicate the number of photos examined, but this does not need to include those in the dark or excluded by human presence.
• It is important to indicate the proportion of the experiment covered by photos; I incorporated that information into my suggested changes to the camera deployment information, but it probably needs correction.
• Clarify how many different individuals checked the photos and whether there was any check for reliability of the photo checks such as missed predators or urchins. (Provide in parentheses the initials of the person who examined the photos if it was only one of the authors and be explicit if others such as research assistants were involved.)
L216ff. I also provided some suggestions for simplifying the descriptions of your codes.
• However, ‘interaction’ needs to be more completely defined or at least a careful description examples of this event because it becomes an important part of your conclusions. Remember that another researcher should be able to repeat your study.
• A very important point is that a single predator visit might result in a large number of coded photos. Thus, the large number of photos in various categories might be misleading. In my experience, most camera trap analyses adopt a criterion to distinguish different occurrences, for example, photos separated by more than 10 min, and doing so would make sense in the present study. Doing so would also allow you to estimate visit durations. If it is not possible to report number of independent visits, I think you should at least indicate whether or not the numbers represent independent visits and discuss possible implications of your interpretation of the effect of interaction number.
L247. Not clear what alive and dead separated by a comma implies. Do you mean a ratio? Wouldn't the response variable be simply number alive? But you have to deal with different starting numbers for caged and uncaged treatments?
L254. Not clear what treatment means? Does treatment mean source as in previous paragraph? It could also refer to caging. Be careful to use terminology consistently and check Methods and Results to confirm that this is so.
L255. Is this really ‘number of times’ or ‘number of photos’, not necessarily the same.
L256. Not clear if you ran a separate analysis for code 3 and code 4 or if they were somehow combined.
Fig. 2 (will become Fig. 3; notes on pdf). The caption is incomplete. Please explain the numbers near each module. Two-headed arrows are more usual to indicate distances between objects.
Fig. 4 Tables, figures and text should not overlap excessively. I think the text is ok because that categorizes the different codes. However, the caption to Fig. 4 and Table 1 overlap excessively. I suggest removing Table 1 and having information only in the Fig. 4 caption. After the illustrations of 6 codes, you can write something like. Codes X (description), Y (description) and Z (description) are not shown. Please note that I have provided detailed suggestions for clearer and more concise descriptions of the codes on the pdf Table 1. Please use these in the figure caption.

Results
Fig. 5. The fonts for legend, axis numbers, labels and symbols are too small to be really clear on reduction to page size. Use a photocopier to reduce the figure to see how it will look on the page. Remove ‘mean’ from the axis label because you show more than the mean. Add zero to the y-axis. The caption should give sample size and units as well as sample sizes and symbols. For example, ‘Mean +/- SE retention (%) of D. antillarum over the 10-day experiment. The four treatments are caged wild D. antillarum (red circles, dot-dash line, n = 8), uncaged wild D antillarum (purple circles, dashed line, n – 16), . . .’
L290. I think these are photos, not sightings.
Table 2. Consider adding percentages.
L303-305. I may be missing something here, but I don’t understand the calculation of these figures. Is this the number of photos (<10 cm) per urchin or the number of photos per reef? Is it calculated over the 10 day period, even when there were no urchins on a reef? Is it the same calculation as L308-309 but without the stats? Even if non-significant the statistics should be presented. It is important the units (per reef, per time) need to be clear. See my suggestions on the pdf for making this more explicit.
L311. This analysis does not determine cause and effect because interactions was not a controlled variable. It is a correlation.

Discussion
L363-367. This section is not sufficiently clear. Are triggerfish territories sufficiently large to encompass the experimental site? Are they interspecific territories? If they involved harems, wouldn’t you have seen more than one individual in the roving survey? Is the seasonality of Saba similar enough to the location studied by Rivera Hernandez that the breeding season is likely to be similar? You should at least acknowledge that it was not in your region.
L376ff. After the paragraph about the shelters being sufficient to prevent predation, you should start a new paragraph addressing the relationship between interactions and overnight loss of urchins. This pattern was important enough for it to become your title so it deserves a more focused, thoughtful and critical discussion. In this paragraph, you do not state as explicitly as in the Abstract that you favour the hypothesis that urchins ‘voluntarily’ left shelters after receiving more attacks and were predated in the open at night. Be explicit and consider the evidence and limitations of this argument. Here are some points to consider. I have some experience with Caribbean fishes, but not with this system, so I may be making wrong assumptions. However, other readers may think along these lines and the suggestions may trigger you to come up with alternative ideas.
• Remember that it is a correlation so causation is an inference that requires care. Anything about particular urchins or shelters that made them more likely to leave or be predated during the night could have provided cues that resulted in triggerfish attacking those urchins/shelters more often.
• It would help to provide any details on what behaviour was involved in the interactions.
• It would help to know how the number of photos related to the number of visits. Were visits more frequent or more persistent or both on the shelters that were empty in the morning? Would the correlation between number of visits or visit duration and absence the next morning show the same correlation?
• You provide statistics to support the correlation between interactions and loss of urchins, but the only quantitative figures is the post-hoc comparison of the mean numbers on interactions for lost and retained urchins. Consider adding a graph showing interactions on the x-axis and proportion abandoned/probability of abandonment on the y-axis.
• Do triggerfish cease to be active while the photo quality is still adequate for analysis or is it possible that the triggerfish returned at dawn, dusk (or overnight if they are not strictly diurnal) and succeeded in an attack when the urchin may have become more active or moved closer to the entrance of the shelter?
• Although you favour the emigration hypothesis, how do you exclude the potential for nocturnal predators that might be attracted to the same individuals or shelters?
L384. Once you start discussing alternative explanations for migration out of shelters, you should start a new paragraph, and make the links to previous assumptions about leaving the shelters clear.
L400. Specify that being in the back of the shelters was based on day-time observations.
L400. Explain why field-collected individuals might not have a normal diurnal rhythm.
L401. You refer to the high number of attacks but never provided the number of attacks, only the number of photos.
L401-403. I don’t understand this explanation at all. It seems to be a circular argument in explaining more attacks on the basis of more attacks. Please clarify or remove.
L403. The size difference is the only one you present in the abstract. If this is the one you favour, shouldn’t it go earlier in the series of potential explanations?
L406. Are the shelters tapered? This wasn’t explained in Methods. If not, why might larger individuals not be able to retreat as far?
L409ff. This paragraph returns to the reasons for leaving shelters and then goes on to nocturnal loss to predation (L419). The material needs to be better organized (see my notes re L375 and 384).

References
Please check the references meticulously. There are odd breaks, inconsistent margins, capitalization of journal article titles, inconsistent capitalization of journal titles, and even incomplete references. I highlighted a few of these, but you need to check them all.

·

Basic reporting

A few minor editorial comments have been made with the intent to improve clarity. Likewise, some suggestions in terms of new, relevant literature references as well as opinions on some slight modification of results interpretation have been included. Well-curated and complete dataset is shared, including formatting for independent statistical analysis.

Experimental design

Solid experimental design, with a couple of minor comments on ways the authors may address the inevitable imperfections.

Validity of the findings

Similar to Basic Reporting, the underlying data are provided in a excellent format, some minor suggested comments are provided on the way results are interpreted in the discussion.

Additional comments

Overall a well-designed and executed experiment. The authors have framed their results well within the context of the existing literature and generally do a good job of acknowledging areas of the experiment that may confound interpretation of results. The line-referenced comments below are suggestions or questions for consideration that I believe would improve the manuscript and render it suitable for publication. They are relatively minor and, in my opinion, should the authors do a satisfactory job of addressing them this manuscript will be suitable for publication.

Specific comments:

L50 – delete “the”

L51 – change “will” to “can”

L61 – suggest “…often attached during the day frequently vacated their shelters…”

L80-81 – Is this not at the high range of densities that were observed? Average densities were closer to 4-5/m2, correct? This sentence reads as if that is the range of densities one would expect to see on any given Caribbean reef before the die off.

L95 – delete “In 2016”

L109-110 – For this sentence and the following paragraph, also see Pilnick et al. (2023) Long-term retention and density-dependent herbivory from Diadema antillarum following translocation onto a reef restoration site. Coral Reefs 42, 629-634. Also applicable to Discussion L385-386.

L133-135 – Suggest also framing the comparison between lab-reared and wild as an aim of the study. The previous couple of sentences read as if the only aim is predator identification.

L136 and elsewhere (e.g. Discussion L396-401) – see recent update to Sharp et al. (2018); Sharp et al. (2023) Diurnal sheltering behavior of hatchery-propagated long-spined sea urchins (Diadema antillarum): a re-examination following husbandry refinements. Bulletin of Marine Science 99, 97-108

L144 – “South” should not be capitalized here.

L170-172 – What was the timeframe between collection of the wild urchins and stocking onto the reef modules? If they were held in artificial conditions for any period, might this have affected behavior?

L193 – Suggest specifying that this is “per camera deployment”, I read it initially as inferring multiple Diadema deployments.

L247-248 – The abstract suggests that urchin mortality was not directly observed, how can dead urchins be a response variable? Should it be “urchins missing” or something similar?

L269-270 – The sentence beginning “To account for dependency…” should be revised for grammar.

L284 – Not sure about using the word “survival” here if they were not confirmed as predated or otherwise dead.

L342 – There is some confusion here as the previous paragraph cites de Breuyn 2021 as having 60% retention after 2 months. I assume that “removal” here refers to some proportion of the restocked urchins within hours?

L341-367 – I wonder if here or elsewhere in the Discussion the authors might consider the depth of the experimental site as a factor potentially contributing to differences in what was observed versus may have been expected? Without thoroughly going into the referenced literature, other than de Breuyn 2021, I would be surprised if many other Diadema restocking studies have taken place at depths equal or greater than 19m.

L403-407 – Because observation at night was not possible, would the authors consider acknowledging that there may have been movement among reef modules during that time (when urchins would already be expected to be more mobile)? On a related note, as urchins were placed in the camera-facing shelters, during diver surveys, were urchins ever observed in shelters on the opposite side of the reef module? If not, this would support a lack of movement among modules. Are there any other means by which nighttime movements could potentially be inferred?

L409-413 – In my opinion, the data provide strong evidence that attempted predation contributed to Diadema emigration and the authors do state that food searching cannot be totally disregarded. However, I would go so far as to include nocturnal movements for feeding purposes in the same vein as movement for aggregation is treated (L391-393) by more strongly acknowledging it as a possibility.

L418-421 – Given the relatively small size of these urchins and without firsthand knowledge of the surrounding patch reefs, I would suggest that the possibility they were present but not seen during diurnal opportunistic surveys could be relatively high – depending on the rugosity and type of crevice structure available. Figure 2 is helpful, but I’m curious what the closest and average distances were from reef modules to adjacent patch reef.

Figure 4 caption – Some of the panels do not have the corresponding code listed. Also, consider labelling the panels with the corresponding Code number directly, rather than a-f and then having Code numbers in the caption.

Figure 4 – I realize it’s necessary to somehow assign a cutoff, but how can you be certain that the fish in panel b is within 50cm of the shelter? I imagine this would apply to any photos in which the predator is directly between the camera and the reef module.

Reviewer 2 ·

Basic reporting

This was an informative and well-researched manuscript about a field experiment to assess whether wild or captive raised Diadema had different survival rates on artificial reefs. The experiment was able to test the a priori hypothesis that daytime predation pressure was an important reason for declines in Diadema densities after restocking to artificial reefs using timelapse underwater photography and diver surveys. I found the manuscript to be very well-written, solid in its basic reporting, with many references that were appropriately cited, and I think the figures and tables are useful and informative. I have a few suggested areas that need clarification, explained below.

Experimental design

The experimental design and methods seemed appropriate to this reader. The research question was well defined and meaningful, with methods generally appropriate. I am not a coral reef biologist, so I am not familiar with the particular players in this study system, but do I work extensively with sea urchins, and am very familiar with experimental subtidal ecology. I think the study experimental design would have been better if there could have been a minimum three replicates for all treatments, and I think it would have been possible to deal with this issue even though there were only 4 cameras (see my next comment). That said, I did think the experimental design accomplished what needed to be done and I think the study ends up making a helpful contribution to the state of knowledge.

Validity of the findings

I think the findings presented, given the methods employed and evidence used, are valid and interesting. The only real logical flaw from my perspective was that even though there could only be two replicates that the 4 cameras could ‘see’, there could have been additional replicates that the cameras did not see which would have bolstered the level of replication for quantifying changes in density etc. Thus, using the limited cameras as an excuse for the limited replicates is not too convincing to me. But, that said, I don’t think this criticism is a big enough deal to seriously challenge this study, and I think that the authors have made good progress with this study in ultimately helping to answer (or get closer to the answer) of what is going on with these urchins after restocking efforts.

In addition, it would be good if the authors can clarify what they mean on line 229 when they say they are not sure if they double counted code 1s. This could be problematic, and from my perspective, I cannot see why there would indeed be double counts (see my comment #4 below).

Additional comments

The manuscript is pretty solid. I only have a few possible changes/clarifications I would request.
1. The ms starts in the third person, and then switches to the active voice. I would be consistent and personally prefer the active voice throughout.
2. The ms starts with using future tense, i.e., ‘...will...’ (e.g., lines 51, 137) but then switches to past tense for most of the ms. The whole thing was in the past at this point, and it should be consistent.
3. On line 128 what is meant by ‘because spines with tissue chunks were observed’? Can the authors clarify? Are they saying that they saw photographic evidence or divers recorded remains of urchins (spines with tissue chunks) which are consistent with predation? The wording currently is a little vague here.
4. Line 229-231. Authors state that they cannot exclude the possibility that they double counted code 1s. I appreciate the honesty of the authors but also don’t follow why they would have double counted these. If this is a possibility, I would encourage them to double check and/or re-code the analyses so that this is not happening. If it is happening, then can they take appropriate steps during the analysis to ensure that this is not biasing their findings?
5. Line 313-314. I was confused by this sentence about the shelters retaining at the end of the night and the connection to the interactions with predators on the previous day. I think what was done here is ok but the wording was awkward for me, I had to read it several times and I am still not sure if the authors are saying what they mean here? I think it gets clarified in lines 377-379 in the discussion, but it was still hard to understand in the results.

---

## Round 0.2 · accepted · Accept

The authors have done a good job of revising and explaining their revision. I believe it is now ready for publication with the addition of a couple of minor changes. I have confirmed these with the corresponding author already to be sure they are acceptable.

Minor changes to be made:
L219 specify that ‘sightings’ refer to individual photos ‘Predator sightings in individual photos were coded 1 – 7 . . .’
L260 Need to clarify definition of independent visits: ‘Because photos were taken only 5 s apart, a predator usually appeared in a sequence of multiple photos. We considered visits as independent only if they were separated from other photos of a conspecific predator by at least 10 min.’
L318-320 Queen triggerfish visits were more frequent in the first half of the experiment, where 1.0 - 1.8 visits per hour were recorded (Table 2). In the second half of the experiment, this decreased to 0.4 - 1.0 visits per hour.
L320 Mean duration per queen triggerfish visit varied among days and was highest on the first day (11 ± 15 minutes) and lowest on day 8 (0 ± 1 minute).
L358 ‘majority were’ (majority is plural here because it refers to multiple individuals)
Fig 4 caption (last line) Photos, not pictures
Table 2. Second column. Replace heading by ‘Mean independent visits per hour’. Replace commas by periods in the numbers in this column


Note the favorable comments on the reviews they received, which might be useful for publicity for the journal.